# The Relevance of Time in Biological Scaling

**DOI:** 10.3390/biology12081084

**Published:** 2023-08-03

**Authors:** Douglas S. Glazier

**Affiliations:** Department of Biology, Juniata College, Huntingdon, PA 16652, USA; glazier@juniata.edu

**Keywords:** scaling with size (allometry) and time (allochrony), biological rates and durations, physical dimensions of space and time, life histories, mortality, evolution

## Abstract

**Simple Summary:**

Large organisms are not simply proportionately magnified versions of smaller related organisms. The magnitude of various features of organisms often changes disproportionately (allometrically) with increasing body size, thus causing fundamental shifts in body shape and function. These “biological scaling” patterns, especially for the rates and durations of various living activities, have traditionally been explained as being driven by body-size-related changes in the rate of metabolism or energy use. Here, I explore whether a “time perspective” may help explain biological scaling patterns as much as or even more than that of an “energy perspective”. After identifying problems with existing time perspectives based on simple universal “four-dimensional space-time” and “biological clock” concepts, I advocate further investigation of three other new or unappreciated time perspectives. They include (1) biological scaling based on time (allochrony) rather than size (allometry) and viewing the body-size scaling of the “pace of life” in relation to (2) fundamental time limits set by the “pace of death” and (3) evolutionary patterns of the origination/extinction of large-bodied species during geological “deep” time. These new or neglected time perspectives have the potential to revolutionize our understanding of biological scaling and its applications.

**Abstract:**

Various phenotypic traits relate to the size of a living system in regular but often disproportionate (allometric) ways. These “biological scaling” relationships have been studied by biologists for over a century, but their causes remain hotly debated. Here, I focus on the patterns and possible causes of the body-mass scaling of the rates/durations of various biological processes and life-history events, i.e., the “pace of life”. Many biologists have regarded the rate of metabolism or energy use as the master driver of the “pace of life” and its scaling with body size. Although this “energy perspective” has provided valuable insight, here I argue that a “time perspective” may be equally or even more important. I evaluate various major ways that time may be relevant in biological scaling, including as (1) an independent “fourth dimension” in biological dimensional analyses, (2) a universal “biological clock” that synchronizes various biological rates/durations, (3) a scaling method that uses various biological time periods (allochrony) as scaling metrics, rather than various measures of physical size (allometry), as traditionally performed, (4) an ultimate body-size-related constraint on the rates/timing of biological processes/events that is set by the inevitability of death, and (5) a geological “deep time” approach for viewing the evolution of biological scaling patterns. Although previously proposed universal four-dimensional space-time and “biological clock” views of biological scaling are problematic, novel approaches using allochronic analyses and time perspectives based on size-related rates of individual mortality and species origination/extinction may provide new valuable insights.

## 1. Introduction

Biological scaling has long been a subject of major interest in comparative biology and ecology. This field of study focuses on how various biological traits relate to the size of a living system, which can often be quantified by a simple power function, such as
*B* = *aS^b^*,(1)
where *B* is the magnitude of a biological trait, *a* is the scaling coefficient (antilog of the intercept in a log–log plot), *S* is the size (e.g., mass) of a living system (e.g., cell size, organ size, body size, colony size, etc.), and *b* is the scaling exponent (loglinear slope or scaling “power”) [1,2,3,4,5,6]. Trait sizes (*B*) often vary disproportionately (allometrically) with *S* (*b* ≠ 1) in various ways for reasons that have been much debated. Although many biological scaling relationships have been historically explained in terms of universal physical laws resulting from single deterministic causes (e.g., the 2/3-power “surface law”, based on the Euclidean geometry of surface area and volume; or the 3/4-power law based on the fractal or fractal-like geometry of resource-transport networks), many investigators now advocate holistic, contingent multi-mechanistic models that better explain the diversity of allometric exponents actually seen. This is especially true for the body-mass scaling of metabolic rate, where *b* varies considerably from near 0 to >1, both within and across species and clades [7,8,9]. Evidence has been steadily growing that metabolic scaling exponents are phenotypically plastic and evolutionarily malleable and not the simple result of physical constraints (e.g., [6,7,8,9,10,11,12,13,14,15,16,17,18,19]). This does not mean that physical constraints, such as those related to surface area/volume ratios, branching resource-transport networks, and finite space and time limits, play no role at all in biological scaling. The evolution and phenotypic expression of biological scaling must occur within the context of the physical properties of living systems. However, in many cases, physical constraints may act more as boundary constraints on the range of possible scaling exponents rather than as deterministic causes of specific exponents [7,9,11,20].

Considerable attention has been given to the nature and causes of metabolic scaling, principally because all biological activities are energized by metabolism (i.e., the collection of biochemical reactions that transform environmental resources into various biological structures and activities). Indeed, metabolic rate has often been considered to be the “pacemaker” for various biological and ecological processes (e.g., [21,22,23,24,25,26,27,28,29]), a view that has been supported by the frequent observation that the body-mass scaling exponents for the rates and durations of various developmental, physiological, behavioral, and ecological processes parallel that for metabolic rate, or nearly so (but see [28,30,31,32,33,34,35]). In short, many investigators have regarded metabolic scaling as being the driver for the scaling of many other biological traits. However, correlations between the body-mass scaling exponents for various biological processes do not definitively inform us of the nature or direction of the cause-and-effect mechanisms involved [2]. For example, the scaling of metabolic rate may drive growth rate or growth rate may drive metabolic rate, or both may be co-adjusted or mediated by a third factor (e.g., body size, resource availability, mortality rate, etc.) (see, e.g., [10,12,28,34,35,36,37]). The “master driver” of various biological scaling patterns may not necessarily be metabolic rate but rather other important body-size-related limiting factors such as resource availability, resource-acquisition ability, resource-allocation priorities, body-space limits, and (or) mortality-imposed time limits, all of which should be considered when attempting to develop a comprehensive theory of biological scaling.

In this review, I focus on the relevance of time in biological scaling. First, I make the obvious point that time is intrinsically involved in the rates and durations of various biological processes and, thus, their scaling with body size. Second, I critically evaluate whether considering time as an independent fourth dimension, in addition to the three dimensions of space, as is fundamental for understanding the dynamics of physical systems [38,39,40], is equally useful for living systems, thereby, in particular, explaining the frequently observed, near quarter-power scaling of many biological rates and durations. Third, I argue that it may be profitable to scale biological traits, not only in relation to physical size (allometry) but also in relation to biological time (allochrony). Fourth, I hypothesize that the tempo/timing of various biological processes/events, i.e., various indicators of the “pace of life”, may not necessarily be set by the rate of metabolism, i.e., the “fire of life”, but rather by mortality-imposed time schedules, i.e., the “pace of death”. In doing so, I argue that the body-size scaling of mortality rate imposes body-size-related limits on the available time for performing various biological activities, thereby acting as the “master driver” of the scaling of many biological rates and durations (though reciprocal causation may also occur). Fifth, I speculate about whether our understanding of biological scaling may also be improved by considering how various scaling patterns may have evolved in geological time through the selective origination and extinction of large-bodied organisms.

As a result, I conclude that, although a simple geometric space-time perspective should be abandoned in the field of biological scaling, a mortality-based “biological time perspective” and a macroevolutionary “geological time perspective” may provide fundamentally new insights into the causes of many biological scaling patterns seen in the living world. These new time perspectives, in addition to underused methodologies involving the scaling of various biological traits in relation to both system size and temporal duration, may have many theoretical and practical benefits.

## 2. Major Ways That Time May Be Relevant in Biological Scaling

As noted in Section 1, many biological scaling patterns have been explained as being the result of the body-mass scaling of rates of metabolism or energy use (e.g., [21,23,26,41,42,43,44,45]). Although this “energy perspective” may be useful in many cases, here, I argue that a “time perspective” may be equally or even more useful, especially with regard to the scaling of the rates and durations of various biological processes/events. In doing so, I review and evaluate various ways that time may be critically involved in biological scaling.

### 2.1. Time and Scaling of Biological Rates and Durations

Biological rates and durations include time by definition. In a physical (or metaphysical) sense, they can be considered four-dimensional, involving a physical time dimension in addition to the three dimensions of physical space (cf. [46,47,48]). Organisms constitute space-time processes that are ever-changing and thus cannot be completely described in only three dimensions. However, the question remains whether a “4D space-time” perspective provides any significant insight into the body-size scaling of biological rates and durations, a topic considered next.

### 2.2. Is Time a Legitimate Fourth Dimension in Biological Scaling?

Some scientists have suggested that quarter-power scaling, as often observed for various biological rates and durations in relation to body mass, arises because of the inherent 4D space-time nature of living systems (see Section 2.1 and [3,49,50,51,52,53,54,55,56]). This view assumes that biological time is a “universal clock” [57] that represents an independent fourth dimension commensurate with the three dimensions of space. Below, I criticize this view by making three major points: (1) biological rates and durations often do not follow quarter-power scaling, (2) biological time covaries with system size and thus cannot be considered a fourth dimension that is independent of the three dimensions of space, and (3) biological time also varies with age, temperature, and type of process, organ, or tissue. In Section 2.3, I further describe how various biological time durations vary discordantly with one another, thus contradicting the “universal biological clock” view. 

#### 2.2.1. The Scaling of Biological Rates and Durations Are Diverse and Do Not Necessarily Follow Quarter-Power Scaling

Although the body-mass scaling of the rates and durations of various biological processes often show quarter-power relationships, at least approximately [2,3,4,42,51,56,57,58,59], the extensive literature showing substantial diversity in scaling exponents continues to be underappreciated by many investigators, especially theoreticians and non-specialists. For example, hundreds of studies have shown that the scaling of metabolic rate, a major indicator of the “pace of life”, varies significantly both within and across species, with exponents varying from 0 to >1, thus clearly invalidating a universal 3/4-power law (see, e.g., [7,9,10,20,60,61,62,63,64]. Even mean exponents (“central tendencies”) for various intra- and interspecific metabolic scaling relationships vary substantially (usually between approximately 2/3 and 1) among major taxa of plants and animals [9]. In addition, numerous metabolic scaling relationships are nonlinear in log–log space, showing scaling exponents that vary significantly across different body-mass intervals. Intraspecific ontogenetic metabolic scaling relationships are often curvilinear or multiphasic in many kinds of animals and plants [7,65,66,67,68,69,70,71,72]. Nonlinear interspecific metabolic scaling relationships have also been reported for crustaceans [65], mammals [73,74,75,76,77], terrestrial invertebrates [78], and seed plants [79,80].

Furthermore, several studies have reported scaling exponents for various biological time periods, including gestation time, lactation (weaning) time, incubation time, age at maturity (first reproduction), and life span, that are significantly different from 1/4 [2,20,23,30,32,33,64,81,82,83,84,85,86,87,88,89,90,91,92] or are curvilinear in log–log space [93,94]. In addition, mammalian life span shows a triangular pattern of covariation with body mass that is not adequately described by a single power function [64,84,87,90]. An up-to-date review describing the diversity of body-mass scaling of biological time periods is much needed to counter the persistent belief that this scaling universally obeys a quarter-power law, or nearly so [43,56,95].

The extensive variation in biological scaling patterns described above is not easily explained in terms of a simple four-dimensional space-time view without significant modification (also see [6,96]). In particular, the 4D view of Ginzburg and Damuth [54] (also see Section 2.2.2) predicts that organisms that grow mainly in one or two spatial dimensions should show metabolic scaling exponents near 1/2 and 2/3, respectively, which is contradicted by reports that *b* approaches 1 in several species of pelagic invertebrates that grow chiefly by elongation or flattening [70,97,98]. Moreover, in general, a simple 4D view cannot explain *b* values greater than 3/4 for rates, or less than 1/4 for durations [6].

#### 2.2.2. Biological Time Is Not an Independent Fourth Dimension Commensurate with Spatial Dimensions

Several investigators have suggested that the key to understanding quarter-power scaling of biological rates and durations is to consider organisms as four-dimensional systems [99,100] with time as the fourth dimension [3,49,50,51,52,53,54,55,56]. According to the specific 4D space-time view of Ginzburg and Damuth [54], the 3/4-power scaling exponent results from the rate of resource supply for a biological process being a function of the three dimensions of 2D surface area and 1D time, whereas the rate of resource use is a function of the four dimensions of 3D volume and 1D time (also see [51,101]). Hence, time is considered to be an independent dimension that is commensurate (proportionate) with each of the spatial dimensions of length, width, and height [49,54,96,102,103,104]. In addition, it has been suggested that a key time period in biological scaling is generation time [54,55,56], also often called “generation length” (e.g., [105,106,107,108]), which appears to be an implicit or inadvertent acceptance of time as being equivalent to spatial length.

However, unlike physical (astronomical) time, biological (physiological) time is not an independent dimension equivalent to each of the three spatial dimensions of length, width, and height. Physical time proceeds independently of an organism’s properties and activities. It can be measured by the frequencies of cyclic astronomical events, such as the rise and fall of the sun in the sky and the changing of the seasons, resulting from the earth rotating around its axis and revolving around the sun, both of which occur independently of organismal size. By contrast, biological (physiological) time clearly depends on an organism’s spatial dimensions, activities, and body temperature [3,4,46,59,101,103,109,110,111,112,113,114,115]. It can be measured by the frequencies of cyclic cellular, developmental, and physiological events that all scale with organismal size, typically proceeding faster in small vs. large organisms [3,4,42,51,52,104].

Therefore, biological time should not be considered an autonomous (self-standing) fourth dimension because it is not independent of or proportionate (isometric in a 1:1 way) with an organism’s three spatial dimensions. To test this claim, I calculated least squares regressions of log_10_ generation time (G) in relation to log_10_ body length (L), using data for unicellular and multicellular organisms compiled by Bonner [116] in his classic book “Size and Cycle” (note that log-transformation, as often used in scaling analyses, permits proportional relationships to be readily discerned: see [117]). I focus on G because, as noted above, it has recently been singled out as a key fourth dimension in the body-size scaling of the rates and durations of various biological processes [54,55,56]. My claim is supported by the observation that G (using age at first reproduction as a proxy) scales disproportionately (allometrically) with L (Figure 1) according to a scaling exponent *b* = 0.804 ± 0.079 (95% confidence intervals; *N* = 46), which is significantly different from 1. Allometric scaling relationships also occur for unicells (*b* = 0.731 ± 0.249; *N* = 10) and animals (*b* = 0.607 ± 0.194; *N* = 30), each analyzed separately, and thus, hypometric scaling of G in relation to L (*b* < 1) is not a phylogenetic artifact. Other datasets for specific animal taxa show hypometric scaling of G (i.e., generation time or age at maturity or first reproduction) with L, as well (Table 1). Of the 10 datasets analyzed here, all show scaling slopes that are <1, nine significantly so (Table 1). In addition, *b* for age at adult maturity (developmental time) in relation to L is significantly less than 1 (0.425 ± 0.307; *N* = 82) for orthopteran insects [118]. Note that I did not include phylogenetic effects in the scaling relationships calculated in Table 1 because I merely wanted to test whether the scaling slope was less than one, which is almost always observed regardless of the taxon. Including phylogenetic effects would unlikely change this general conclusion but may alter the exact value of the exponent for each taxon. In addition, I used least squares regression (LSR) analyses rather than reduced major axis analyses because body length tends to be measured with less error than generation time, thus making LSR more appropriate (also see Section 2.3).

Interestingly, dimensional analysis predicts that if G is proportional to L^0.804^, as occurs for my analysis of the data compiled by Bonner [116], and assuming that body mass (M) is proportional to L^3^ (as occurs in isomorphic organisms), then G should be proportional to M^0.804/3^ = M^0.268^, which is near M^0.25^, as predicted by the 4D theory of Ginzburg and Damuth [54], which assumes that G is an independent dimension proportional to L^1^. Similar results occur for unicells (M^0.731/3^ = M^0.244^) and animals (M^0.607/3^ = M^0.202^) analyzed separately, though dimensional analyses of the scaling of G (i.e., generation time, age at maturity, or age at first reproduction) with M in specific animal taxa show variable results (e.g., cladocerans: M^0.479/3^ = M^0.160^; orthopterans: M^0.425/3^ = M^0.142^; teleost fish: M^0.799/3^ = M^0.266^; squamates: M^0.313/3^ = M^0.104^; reptiles: M^0.292/3^ = M^0.097^; birds: M^0.694/3^ = M^0.231^; and mammals: M^0.816/3^ = M^0.272^, or M^0.816/3^ = M^0.225^).

Regardless, the 4D theory of Ginzburg and Damuth [54] is incorrect both empirically and logically. It is incorrect empirically because G is usually not proportional to L^1^, as just shown. It is also incorrect logically, because, if G were a truly independent fourth dimension proportional to L^1^, G should scale as M^1/3^, not as M^1/4^, as recognized by Lambert and Teissier [102], whose pioneering dimensional analysis assumed that biological time T should scale generally as L^1^. For all of the dimensional analyses described above, calculated scaling exponents for G with M are <1/3. Therefore, an explanation of why G (and T more generally) should scale as M^1/4^ does not follow from simple 4D space-time theory, because T is not an autonomous dimension that is commensurate (proportionate) with L but rather a size-dependent variable that scales allometrically with L (also see [96]).

Accordingly, a realistic explanation of the size scaling of G and T does not appear to depend on 4D space-time theory but rather on other factors that cause the rates and durations of various biological processes to scale with L. As explained in Section 2.4, these factors may include harmful mortality-causing environmental hazards whose overall impact scale with L and M, being more severe in small, relatively vulnerable organisms compared to larger, less vulnerable organisms (also see [18,35,125]).

One may also question whether spatial length, width, and height should be considered independent proportionate dimensions when examining biological scaling relationships in living systems. This is true only for “isomorphic” organisms that have the same body shape regardless of body size. In non-isomorphic organisms, spatial dimensions may not be proportionate with one another (e.g., [126]). As length increases, width and height may change disproportionately if an organism grows by elongating, flattening, or thickening its body. As a result, M often does not scale in proportion to L^3^ but with exponents greater or less than 3 (see, e.g., [70,71,97,98,127,128]). This fact further weakens the general usefulness of simple 4D theory in biological scaling.

#### 2.2.3. Biological Time Does Not Follow a “Universal Clock” but Varies with Body Size, Age, Temperature, and Type of Process, Organ, or Tissue

Several biologists have suggested that the rates and durations of various biological processes are synchronized by an internal biological clock or “periodengeber”, largely based on the claim that various biological time periods scale similarly with body mass (often to the 1/4-power) [3,51,56,104,114,129,130,131,132,133,134,135]. However, this synchronized “universal biological clock” view has five major problems. First, this view could be considered “not an explanation at all but rather just a renaming of an empirical phenomenon” ([125] p. 196, but see [132,133,134,135]). Second, many kinds of biological event frequencies or durations show variable scaling that does not necessarily follow a 1/4-power law (see Section 2.2.1). Third, even if quarter-power scaling were universal, at least approximately, the conclusion that all or most biological time periods vary in a parallel way, or nearly so, among populations, species, and higher taxa does not necessarily follow. This is because variation in biological time periods does not depend entirely on variation in body size: a significant proportion of this variation may be unrelated to variation in body size. Indeed, “residual” variation that is orthogonal to a body-size regression can be substantial. For example, life span can vary by over 10-fold in mammals of equivalent size (see Figure 1 in [87] or Figure 2 in [90]). Fourth, different biological rate processes or time periods may vary discordantly in response or in relation to a variety of intrinsic (biological) and extrinsic (environmental) factors, as discussed further in this section and Section 2.2.2. Fifth, correlation analyses of various biological time periods reveal that they may vary disproportionately with one another in an “allochronic” or “heterochronic” way [136,137,138] (see Section 2.3), even if they show parallel or nearly parallel scaling (allometry) with body size. In short, parallel allometric relationships do not necessarily mean that biological time periods covary proportionately (in an “isochronic way”), as if they followed the same clock (also see Section 4).

Biological time does not proceed with a uniform, consistent rate (like physical time), even within the same species but varies with the biological process, organ, and tissue being considered. Although the harmonious, synergistic action of multiple biological processes may have adaptive value [139], they may also show substantial “dissociability” in response to a variety of intrinsic and extrinsic factors [28,140]. Different biological rate processes or time periods may vary discordantly with body size [32,33,118,141,142], age [110,136,143,144], genotype [145], temperature [28,103,110,112,113,143,146,147,148,149,150,151,152,153,154,155,156], and other biological and environmental factors [28,140,157]. Box 1 provides specific examples.

In summary, organisms are dynamic “temporal mosaics”, including multiple processes running at different tempos at various hierarchical (cell, tissue, organ, and organism) levels, depending on various intrinsic and extrinsic factors. Various intrinsic and extrinsic factors may also interactively cause discordant variation in the rates or timing of various biological processes. For example, temperature effects may vary with body size [28,63,146,158,159], age [143], type of process (see, e.g., [110,143,154]), and other factors [147]. As Carraco et al. [160] have remarked, with respect to embryonic development, different species and tissues dance “to a different beat”.

Box 1Examples of dissociation between rates or durations of various biological processes.  Conventional belief is that the rates and durations of various biological processes of organisms are synchronized (see references in text). However, many examples of biological rates or durations being influenced discordantly by various intrinsic and extrinsic factors exist. Some selected examples include:
(1)Avian incubation and fledgling periods scale differently with body mass [141], as do the durations of gestation and lactation in primates [142], gestation time, weaning time, age at first reproduction, and life span in marsupials [92], gestation time and life span in mammals [32], and life span and age at maturity in birds, mammals, and orthopteran insects [33,118].(2)Genetic and hormonal influences and various environmental factors can dissociate the rates and timing of metabolism, growth, maturation and (or) life expectancy [28,140,145]. For example, changes in temperature can dissociate the rates of growth and maturation, thus causing the well-known temperature-size rule in ectotherms [150,151,155].(3)Differences in age-specific mortality, as caused by artificial selection, can produce changes in various life-history traits, such as growth rate and developmental time, without associated changes in metabolic rate (e.g., [161]).(4)The developmental growth rates of different organs or structures are often unequal (e.g., [1,136,162]). These differences appear to be the result of multiple local regulatory mechanisms [28,163,164,165], and they can be accentuated by experimental manipulation or artificial selection experiments (e.g., [166,167,168]). Disproportionate or discordant variation in the rates and timing of the growth and development of various parts of an organism is so common that it has been recognized by widely used specific biological terms, such as “allometry”, “relative growth”, and “heterochrony”, and has been reviewed in major synthetic books [1,136,144].(5)Cell replication rates or frequencies vary greatly among tissue types, from relatively high in tissues of the skin, blood, lymph, and gastrointestinal tract, which exhibit high levels of “cell renewal”, to relatively low in nervous, sensory and cardiac muscle tissues where no cell renewal occurs (e.g., [169,170,171,172,173,174]). Where cell renewal occurs, cell turnover times vary greatly from hours to days to months [170,174].(6)Given their various levels of cellular activity, it is not surprising that the metabolic rate of various tissue types also varies greatly from relatively high in brain, liver, heart, and kidney tissues to relatively low in adipose and musculoskeletal tissues [4,15,28,66,175,176]. The metabolic rate of skeletal muscle may also change dramatically between resting and active states [177], thereby substantially altering how whole-body metabolic rate scales with body mass [7,11,20,178].(7)The turnover times of various cellular metabolites can vary by over three orders of magnitude in the same organism (e.g., 0.01 to 40 s in *Arabidopsis* plants: [174]).


### 2.3. Scaling Biological Traits in Relation to Biological Time (Allochrony) Rather Than Physical Size (Allometry)

The physical existence of living systems extends in both space and time (also see Section 2.1). Organisms have been defined as “spatiotemporally localized entities” [179,180]. Therefore, I argue that explaining variation in biological traits may be facilitated by not only scaling their magnitude in relation to the size or spatial dimensions of a living system, as is common practice [2,3,4,9,125] but also in relation to its temporal duration (persistence).

Although many interspecific comparisons have revealed that various biological time periods correlate positively with each other (e.g., longevity and age at sexual maturity [84,89,106,181,182,183,184,185,186,187,188,189,190,191,192]), these relationships are often disproportionate (“allochronic”), contrary to a “universal clock” view (also see Section 2.2.3). Smith [137] first used the term “allochrony” to describe how different time periods in the life histories of various organisms or species relate to one other (also see Section 4 for a description of other uses of the term “allochrony”). These relationships may be “isochronic” (showing a slope of 1 in log–log space) or “allochronic” (showing a slope ≠ 1 in log–log space). In a comparative analysis of primate life histories, Smith [137] reported isochronic relationships among female age at sexual maturity, age at weaning, and time of eruption of the first molar tooth but allochronic relationships for estrus cycle length, gestation length, interbirth interval, female age at first breeding and male age at sexual maturity all in relation to age at weaning or time of eruption of the first molar tooth.

Glazier and Newcomer [138] further developed and tested the concept of allochrony, unaware of Smith’s [137] pioneering essay, using the power function
*L* = *aT^b^*,(2)
where *L* is a life-history trait, and *T* is lifetime or a well-defined portion of lifetime (e.g., age at sexual maturity). They showed that, in mammals, the durations of gestation, lactation, and the juvenile period (post-weaning maturation time) all scaled allochronically with age at first reproduction. The scaling slopes (*b*), with 95% confidence intervals, were 0.795 ± 0.079 for gestation time, 0.768 ± 0.090 for lactation time, and 1.138 ± 0.050 for juvenile time, all significantly different from 1 [138]. These allochronic patterns appear to be robust, as they occur in multiple families and orders, body-size classes, dietary types, and foraging modes. They show that as total maturation time increases, the post-weaning juvenile period takes up an increasing proportion, and the pre-weaning fetal-infancy period a decreasing proportion of maturation time. Therefore, the evolution of increased maturation time in mammals appears to be more related to an increase in the post-weaning juvenile period than to an increase in the pre-weaning fetal-infancy period. Glazier and Newcomer [138] further discussed how the juvenile time period appears to relate to the challenges of developing sufficient foraging and locomotor skills to permit reproduction. As a result, faunivorous and arboreal mammals tend to have proportionately longer juvenile periods than mammals with other modes of feeding and locomotion.

As further examples, the age at sexual maturity or first reproduction scales allochronically (*b* < 1) with total life span in gymnosperm and angiosperm trees, cladoceran crustaceans, amphibians, reptiles, birds, and mammals (Table 2, Figure 2). All of the least squares regression (LSR) scaling slopes are significantly less than 1. LSR assumes that the X variable is measured without error. Unfortunately, the exact amount of measurement error for the X and Y variables is unknown. As a precaution, I also calculated reduced major axis (RMA) slopes (= LSR slope/*r*), which assume that the Y and X variables were measured with equal error [193,194,195]. As shown in Table 2, five of the eight RMA slopes are still less than 1. Furthermore, for the three taxa that show RMA slopes near 1, birds and mammals show significantly curvilinear relationships (Figure 2E,F), whereas angiosperms show a slightly curvilinear relationship (not shown in Figure 2B). Therefore, linear RMA slopes near 1 are misleading for these cases because, actually, the relationship between age at maturity (or first reproduction) and total life span changes substantially with increasing total life span. The reptile regression is also significantly curvilinear (Figure 2D). For the reptile, bird, and mammal regressions, the instantaneous LSR slope (first derivative) changes from ≤0 to ≥1 as total life span increases.

**Table 2 biology-12-01084-t002:** Statistical parameters for least squares linear regressions between log_10_ age at maturity (AM) or log_10_ first reproduction (AFR) and log_10_ lifespan (L) or log_10_ maximal life span (ML). Bold slope values are significantly different from 1. Reduced major axis slope values are italicized.

Taxon	AM/AFR (Units)	L/ML (Units)	Slope (±95%CI)	Intercept (±95%CI)	r	N	*p*	Source
Gymnosperms	AFR (years)	L (years)	**0.242** *0.622* (±0.176)	0.921 (±0.422)	0.389	46	0.008	[183]
Angiosperms	AFR (years)	L (years)	**0.659** *0.992* (±0.240)	0.050 (±0.527)	0.664	41	<0.0001	[183]
Cladocerans	AFR (days)	L (days)	**0.430** *0.517* (±0.187)	0.179 (±0.285)	0.832	13	<0.0001	[119]
Amphibians	AFR (years)	L (years)	**0.394** *0.779* (±0.230)	0.037 (±0.243)	0.506	37	0.001	[191]
Reptiles	AFR (years)	L (years)	**0.763** *0.894* (±0.160)	−0.331 (±0.224)	0.853	37	<0.0001	[191]
Reptiles	AM (days)	ML (days)	**0.552** *0.805* (±0.077)	1.002 (±0.307)	0.686	223	<0.0001	[122]
Birds	AM (days)	ML (days)	**0.634** *1.003* (±0.045)	0.370 (±0.175)	0.632	1095	<0.0001	[122]
Mammals	AM (days)	ML (days)	**0.764** *1.063* (±0.033)	−0.209 (±0.124)	0.719	1793	<0.0001	[122]

CI = confidence intervals, r = Pearson product–moment correlation, N = sample size, *p* = probability that r is due to chance.

With respect to the above results, it is also important to note that RMA analyses should not necessarily be preferred when X is measured with error. In such cases, LSR analyses need not underestimate scaling slopes [196]. RMA analyses can also be difficult to interpret [195,196,197]. Therefore, Kilmer and Rodríguez [197] prefer LSR analyses over RMA analyses when measurement error is not large, as appears to be true for most of the data analyzed in my study. In addition, I did not include phylogenetic effects in the scaling relationships calculated here because I merely wanted to test whether the scaling slope was lower than one, which is frequently observed regardless of the taxon. Including phylogenetic effects would unlikely change this general conclusion but may alter the exact value of the exponent for each taxon (also see Section 2.2.2).

In any case, the allochronic relationships documented here indicate that prolonged life spans usually involve disproportionate increases in the durations of adult versus pre-adult life-history periods. In addition, regression slopes of adult or total life span in relation to age at sexual maturity vary substantially and allochronically among various plant groups [181,186], vertebrate taxa [89,186,187,190,198], and taxonomic/ecological groups of fishes [184] and mammals [185,192]. Lemaître et al. [32] have also shown that gestation time scales allochronically with longevity in mammals. Detailed explanations, along with other examples of life-history allochrony, will be published elsewhere. This variation indicates that various kinds of biological time periods are often discordant, and may vary independently in response to various intrinsic and extrinsic factors. Indeed, the age at maturity may vary over 10-fold among reptile, bird and mammal species having the same total life span, and vice versa (Figure 2D–F).

### 2.4. Mortality-Imposed Time Limits on the Rates and Durations of Various Biological Processes and Their Scaling with Body Size

The mortality of all organisms causes all biological processes to be time sensitive, which I propose has major importance for the understanding of many kinds of biological scaling relationships. Here, I assume that a quicker “pace of death” favors (by natural selection) a quicker “pace of life”. If the mortality rate in a population of organisms increases, natural selection should favor a quicker rate of reproduction (and overall pace of life) to ensure (1) the evolutionary success (i.e., gene transmission to the next generation) of individual organisms and (2) the long-term persistence (ecological stability) of populations. A balance between death and reproduction is fundamental in both ecology and evolutionary biology, having been recognized for centuries (e.g., [56,125,186,199,200,201,202]). However, although many biologists have embraced the “rate of living theory” that a quicker pace of life (including metabolic rate) causes quicker aging, reduced longevity, and ultimately a higher mortality rate (e.g., [26,56,203,204,205,206]), the general applicability of this view has been questioned (e.g., [28,207,208]). I argue that the opposite causation, where mortality rate drives fitness-enhancing evolution of the rate of reproduction and the overall pace of life (as promoted by [125,199,200,201,207,209,210], and others), has also been significant. Indeed, both types of causation may be reciprocally important.

Nonetheless, I would suggest that mortality rate is more fundamental than metabolic rate for understanding the body-size scaling of the pace of life, as revealed by following the Socratic method with a logical sequence of questions and answers (see Box 2). Knowing the rate or risk of mortality (or destruction) can help predict the pace of life at the population, organism, organ, tissue, and cell levels, which I illustrate with five examples based on empirical data and/or theoretical models (see Box 3).

Box 2Importance of mortality rate as a driver of rates of various biological processes, as revealed using the Socratic method.  The Socratic method uses questions and answers to achieve a better understanding of a specific subject. I employ this method here to show that it is reasonable to assume that mortality rate importantly affects the rates of various biological processes, as mediated by natural selection over evolutionary time. The questions and answers follow:  1st question: Why do organisms metabolize resources more or less quickly?  1st answer: So that they can grow, mature, and reproduce more or less quickly.  2nd question: Why do organisms grow, mature, and reproduce more or less quickly?  2nd answer: So that they can replace themselves before they die more or less quickly, as favored by natural selection.  Therefore, it is reasonable to conclude that the mortality of organisms importantly dictates (via natural selection) their pace of life. Several lines of evidence provided in Box 3 support the potential importance of mortality rate as a driver of biological rates and durations.

Box 3Examples showing that rates of mortality/destruction may drive rates and durations of various biological processes at different levels of biological organization.  Here I provide several examples of how the rates and durations of various biological processes at multiple hierarchical levels of biological organization are associated with rates of mortality or destruction, which may therefore potentially act as drivers of these biological processes, in an ultimate evolutionary sense, a hypothesis requiring further testing.
(1)At the population level, the maximal intrinsic rate of increase of various kinds of unicellular and multicellular organisms scales with body mass with a slope near –1/4 [211] that is similar to the scaling of mortality rate (*b* ~ –1/4: [34]) but not that of mass-specific metabolic rate (*b* ~ 0) [212,213] or whole-body metabolic rate (*b* ~ 1) [34,213]. As expected from the principle of “ecological compensation” [202], the mortality rate of a stable (persistent) population should be balanced by its reproductive rate [56,125,199,200,201,214]. Of course, populations may temporarily increase/decrease in size, but these trends cannot continue indefinitely because of resource limitation or inevitable population extirpation.(2)At the organismal level, larval growth and developmental rates relate positively to the intensity of adult mortality rate in the fruit fly *Drosophila melanogaster* [161], whereas the post-maturational growth rate (and supporting metabolic rate) of the freshwater amphipod *Gammarus minus* relates negatively to the intensity of adult mortality, as caused by size-selective fish predation [12,215]. Growth rates of the plant *Arabidopsis thaliana* are also inversely related to life span (and thus positively with rate of mortality) [216]. These examples are important because they show that although higher rates or risks of mortality often favor increases in the rates of specific biological processes, the opposite may also occur if increased mortality threats involve size-specific predation, thus favoring reduced rates of foraging and growth that decrease the visibility of adults to visually hunting predators. In short, age- and size-related patterns of mortality may have variable effects on age- and size-specific rates of various biological processes. Comparative studies have also shown that organisms with characteristics that reduce mortality (e.g., flight and hibernation) have slower paces of life (e.g., [217,218]).(3)At the organ level, Sibly and Calow [219] developed a theoretical model showing that the risk of mortality may influence the differential allocation of resources to organs and, thus, their varying growth rates during ontogeny. In fact, empirical data show that rates of growth and photosynthesis of plant leaves are inversely related to their life span (or positively with their rate of mortality) both across [220,221] and within species [216].(4)At the tissue level, it is well known that the replication rate of cells (and the need for “cell renewal”) correlates strongly with their mortality (turnover) rate in different tissue types, which is in turn related to their frequency of injury and exposure to environmental hazards [169,170,171,172,173]. As the reproductive and mortality rates of organisms are matched in stable (non-growing or non-declining) ecological populations, so are the reproductive and mortality rates of cells in stable organismal tissues [219].(5)At the cell level, the rate of synthesis of specific kinds of proteins in animals and plants matches their rate of degradation (turnover) [222,223,224,225], a phenomenon called protein homeostasis or “proteostasis” [226]. Again, rates of destruction appear to drive rates of replacement.

According to a mortality-imposed time perspective, higher mortality rates in small, vulnerable organisms, compared to larger, more protected organisms, have driven the evolution of their more rapid paces of life, as is generally observed. Multiple arguments and lines of evidence supporting this hypothesis have been provided by [35], though we still have much to learn. For example, since mortality rate scales negatively with body mass with slopes usually between −0.1 and −0.4 [26,35,205]), the rates and durations of various biological processes should usually scale with slopes between 0.6 and 0.9 (1+ slope for mortality rate) and 0.1 to 0.4 (1− slope for rate process or 0− slope for mortality rate), respectively, as is indeed often observed [2,3,4,5]. The negative body-mass scaling of mortality rate may result from small organisms being more vulnerable to predation, competition, and other harmful environmental hazards than larger organisms [35,125,200,227,228,229,230,231,232]. As noted by Goatley and Bellwood [229], “small animals can quite literally fit in more mouths, and as such, may suffer a greater risk of predation”. Furthermore, considering life as a whole, large organisms tend to have more protective external coverings (e.g., bark, spines, shells, scales, fur, feathers, and other exoskeletal structures) and relatively small surface-area-to-volume ratios that result in much of their interior body being relatively remote from harmful external environmental influences, as compared to small organisms [35]. Higher mortality rates in smaller organisms may, in turn, favor a more rapid pace of life not only evolutionarily via natural selection but also ecologically and physiologically by being associated with relatively low (decimated) population densities that are below the carrying capacity (K) of the environment and, thus, relatively high per capita resource availability (also see [35,227,228]). In short, a rapid pace of life, as observed in small organisms or those living in unstable or ephemeral habitats [35,227,233], ensures reproduction before imminent death, which may be facilitated by a relatively high available resource supply per individual and large surface-area-to-volume ratios that enhance resource uptake (also see [35]).

Smith [137] additionally suggested that the risk of mortality may help explain why long-lived mammals tend to have proportionately shorter ages at maturity. She speculated that this allochronic pattern represents an adaptive way of ensuring reproduction before death in slow-growing organisms. Alternatively, this trend may be the result of adults of long-lived species being better protected against harmful environmental influences than the adults of short-lived species, thus enabling them to have disproportionately longer post-maturational lives (reproductive life spans) that enable more fitness-enhancing breeding events (as supported by data in Table 2). In either case, a mortality perspective appears to be essential for understanding the allochronic relationships of the age of sexual maturity with life span. Cichoń [234] has also developed a model based in part on extrinsic mortality to predict correlations between age at maturity and maximal life span.

### 2.5. Biological Scaling Viewed in Geological “Deep Time”

The body-size range of life has evolved from being relatively narrow (in the beginning, including only tiny unicellular organisms) to eventually becoming, over many millions of years, relatively broad, now spanning both tiny unicellular and huge multicellular organisms [116,235]. Explicit recognition of this macroevolution of body size over geological time may affect our understanding of biological scaling in three important ways. 

First, the expansion of life’s body-size range has not been linear but branching. Consequently, body-size scaling studies of diverse kinds of organisms should consider phylogenetic effects using various phylogenetically informed methods that have now become available and routinely used, not only in comparative biology generally but also in allometric studies specifically (e.g., [14,125,236]).

Second, the evolutionary appearance of relatively large multicellular organisms probably altered the lives of many smaller unicellular and multicellular organisms in varied but little-understood ways that may have affected the biological scaling relationships that we now see in the living world. For example, many multicellular metazoans and protists that evolved early in the history of life may have preyed upon or competed with unicellular organisms, thus increasing their mortality rate, which, in turn, may have favored (by natural selection) a quicker pace of life (as postulated in Section 2.4). These effects may have been repeated endlessly as ever-larger multicellular organisms evolved that preyed upon and competed with smaller unicellular and multicellular organisms. [However, note there may have been two exceptions to this general trend: (1) the origination of large “structural species” (e.g., trees and corals) may have provided physical refuges for various small species that reduced their mortality rates and associated rates of living (indeed, arboreal mammals tend to have slower rates of mortality and living than do terrestrial mammals of equivalent size [87,138,237,238]); and (2) mortality rates may have been reduced in small organisms (e.g., bacteria) that became endosymbionts in larger host organisms, a hypothesis requiring testing]. Accordingly, I suggest that, in general, the macroevolution of body size in “deep time” may have significantly affected the body-size scaling of the pace of life by favoring increased rates of various biological processes in small, vulnerable organisms relative to those of larger, less vulnerable organisms, thus decreasing the size-scaling slopes for various rate processes, while increasing the slopes for various biological time periods. This is a plausible hypothesis that requires testing, though it will be challenging to do so. Perhaps, comparing biological scaling relationships in present-day ecological communities with different size distributions of species (e.g., island versus mainland communities, and extreme versus moderate environments) may be useful in this respect. It would be especially interesting to compare rates of mortality and living in (1) microbes from favorable versus extreme environments, where larger multicellular organisms are present versus absent, respectively, and (2) organisms from communities with and without apex predators, and associated trophic cascade effects. In addition, experimental studies of the evolution of the pace of life and death in microbes or other small organisms in the presence or absence of consumer species or periodic environmental disasters may also be revealing.

Third, although not yet analyzed, macroevolutionary processes such as species selection may have importantly influenced the body-mass scaling of life-history traits, as well as associated physiological (energetic) traits. Consider that large species tend to have relatively small total population sizes compared to related small species (e.g., [239,240]). Therefore, large species should be more vulnerable to extinction than small species, as supported by multiple lines of evidence, including greater frequency of extinction of larger vs. smaller mammal species during the Pleistocene [241,242], higher frequency of extinction-risk indicators in larger mammals [243] and vertebrate animals more generally [244], and the absence or scarcity of large vertebrate species with small geographical ranges [243,244,245,246,247]. Having high rates of individual energy use may further decrease total population size in the presence of a limited resource supply, thus additionally increasing the relative vulnerability of large species to extinction [243,248]. As a result, species selection (which depends on trait-associated extinction rates, as well as speciation rates: [249,250]) should favor large species with relatively low rates of energy use, thus causing the body-mass scaling slope of whole-body energy use to decrease. The observation that no mammals with both large body masses and relatively high metabolic rates have small geographical ranges [243,248] is consistent with this hypothesis.

## 3. Conclusions and Prospects

In my review, I have provided several arguments and lines of evidence supporting the view that biological time should not be considered an independent fourth dimension commensurate with the three spatial dimensions of living systems (Figure 3A). Therefore, I recommend abandoning the use of a simple four-dimensional space-time view (unless there is adequate major modification) to explain biological scaling patterns. Instead, I advocate for using a time perspective in three other major ways to increase our understanding of many kinds of biological scaling patterns (Section 2.3 and Figure 3B,C).

First, I recommend that comparative studies of phenotypic traits should consider scaling their magnitude in relation to not only the size (spatial dimensions) of a living system (allometry) but also its duration (temporal persistence or life span) or that of other important life-history events (allochrony). All living systems have finite spatial and temporal limits, both of which should be considered in biological scaling analyses. Although a few investigators have recognized the potential value of comparing the magnitude of various phenotypic traits to the life span or other important temporal durations (e.g., age at first reproduction) of organisms (e.g., [137,138,214,251]), allochronic approaches in comparative biology have been relatively neglected and are not yet fully developed.

Second, the inevitability of death for all organisms makes time imperative for all biological processes. Greater mortality (destruction) rates often favor a more rapid pace of life (Figure 3B), whether it be higher rates of growth, reproduction, and/or metabolism in small, vulnerable organisms compared to large, less vulnerable organisms, or higher rates of cell replication in tissues exposed to high levels of damage and environmental stressors compared to less vulnerable tissues, or higher rates of synthesis of proteins that suffer relatively high rates of degradation. This biological time perspective, based on the effects of differential rates of mortality or destruction, can explain why the tempo of various biological processes is often not synchronized. Their tempo depends on matching variable rates of mortality or destruction with equivalent rates of reproduction or replacement. Indeed, I would further argue that the stability (homeostasis) of living systems depends on the heterogeneous dynamics (tempos) of at least some of their constituent parts, whether they be the biomolecules, cells, or tissues in an organism or the species in an ecosystem. If the rate of synthesis occurred equally for all proteins in a cell, those proteins suffering low rates of degradation would increase in concentration disproportionately and thus disrupt the composition and effective functioning of a cell. Similarly, if the rate of cell replication occurred equally in all tissues of the body, those tissues with long-lived cells would grow disproportionately and, like cancer, disrupt the composition and adaptive functioning of an organism. Likewise, if the rate of offspring production occurred equally in all species of an ecological community, those species with high survival would increasingly dominate and accordingly disrupt the composition and effective functioning of an ecosystem. Therefore, contrary to conventional belief (see, e.g., [28,139,252,253,254]), temporal disharmony of at least some of the constituent processes of a living system, rather than wholesale harmony, may be necessary for long-term persistence, a hypothesis worth further testing.

In short, mortality imposes a time-sensitive imperative on the pace of life at a variety of levels of biological organization. Furthermore, if all organisms were immortal or suffered mortality at the same rate regardless of their size (spatial dimensions), biological time would no longer be a size-dependent variable. Only then could biological time be considered a truly independent fourth dimension.

Third, a geological time perspective may expand the scope of possible explanations of biological scaling patterns by including phylogenetic and macroevolutionary effects. Biological scaling relationships may diverge along different evolutionary branches (e.g., [13,255]) and may be affected by the macroevolutionary expansion of the range of body sizes exhibited by living organisms, thus altering ecological relationships among them that affect size-related rates of mortality and associated biological processes (Figure 3C). In addition, although allometric scaling patterns are usually explained in terms of physical constraints, adaptive biological regulation, and micro- and macroevolutionary processes, such as natural selection and species selection (Figure 3C), may also have played important, as yet inadequately understood and appreciated roles, as well (also see [9,14,17]).

Increasing recognition of a time perspective, as developed in this review, could stimulate several new lines of research on life histories and other biological processes. First, analyzing biological scaling in relation to time durations (allochrony), in addition to spatial dimensions (allometry) as traditionally performed, could reveal new kinds of scaling patterns that provide novel insight into the ecological and evolutionary mechanisms causing variation in life-history traits. For example, allochronic analyses may challenge current theory on “life-history invariants”, which is based on parallel allometric relationships that ignore life-history variation independent of body size (e.g., [186]). Although two biological time periods may, at least in some cases, show similar scaling with body mass, thus yielding an apparent invariant ratio between these traits (based on allometric dimensional analysis [49,51,52,129,130]), allochronic analyses may reveal that they vary disproportionately (i.e., allochronically, as several examples described in Table 2 and Figure 2 show). I contend that the identification of life-history invariants should be based not only on parallel allometric scaling relationships but also on isochronic relationships between biological time periods (other problems with identifying life-history invariants are discussed by [256,257,258,259,260]).

Second, a mortality-based biological time perspective may provide the impetus for developing a new major approach to biological scaling, namely a “mortality theory of ecology” (MorTE) [35], which could serve as a useful alternative or complementary viewpoint to the currently influential “metabolic theory of ecology” (MTE) [26,44,261]. As noted in Section 1 and Section 2, many biologists have attempted to explain variation in the rates/durations of various biological processes/events, and their scaling with body size, as being driven by the rate of metabolism or energy use. However, although many studies have reported correlations between metabolic rate and various life-history/demographic traits (e.g., [21,22,23,24,25,26,27,29,262], a similar number have also failed to find significant correlations (e.g., [31,207,263,264,265,266]; see review in [35]). Furthermore, the body-mass scaling exponents for various biological processes/durations (e.g., gestation time, life span, age at first reproduction, individual/population growth rates, etc.) often do not match the scaling exponents for metabolic rate [30,32,34,35,86,212,213], contrary to that predicted by the MTE. Given that variation in diverse life-history traits has been linked to variation in mortality rate (e.g., [84,119,125,207,209,267,268,269,270,271,272], as reviewed in [35]), I recommend future research that examines how the body-size scaling of various life-history traits relate to mortality rate and its scaling with body size. After all, mortality rate scales strongly with body size in a variety of organisms ([26,35,205]), and thus, size-related mortality-based time limits should impact the rates and durations of various biological processes and how they scale with body size. Although a MorTE is not yet fully formulated nor recognized on a par with the MTE, several studies have already suggested or shown that a mortality-based time perspective has much potential for increasing our understanding of the body-size scaling of various kinds of life-history/demographic traits (see Box 3 and [7,10,12,35,37,125,215,273,274,275,276]).

A MorTE may also provide a general evolutionary foundation for biological scaling models that invoke time minimization in addition to energy-cost minimization or energy-gain maximization (e.g., [277]). A limited lifetime causes natural selection to favor a hastened pace of life and, accordingly, the time minimization of diverse biological processes, including foraging, resource uptake and delivery, metabolism, growth, maturation, and reproduction (see, e.g., [201,278,279,280,281]).

Third, a geological time perspective may help stimulate research on how biological scaling patterns have evolved. Most current research focuses only on existing biological scaling patterns, which are merely a recent snapshot of millions of years of evolution. A geological “deep time” perspective may advance our understanding of biological scaling in three major ways (also see Section 2.5). First, it may provide a useful phylogenetic perspective on scaling relationships involving diverse species with different degrees of evolutionary relatedness. Second, it may help explain how the evolution of new species and their resulting new ecological interactions with other already existing species have contributed to the origin/modification of biological scaling patterns. Fundamentally, biological scaling would not be perceived without the evolution of organisms with widely varying body sizes over geological time. The evolution of relatively large predators, competitors, or refuge-supplying hosts and “structural species” (e.g., trees and corals) may significantly increase/decrease the rates of death and living of smaller species affected by them. According to this view, biological scaling patterns are not merely the result of physical constraints but are ecologically sensitive and ever-evolving as the body-size spectrum of life changes. Third, a geological time perspective may stimulate research on how macroevolutionary processes such as selection (sorting) at the population and species levels [249] have affected biological scaling patterns, a topic that, to my knowledge, remains unexplored.

In conclusion, although simple universal “four-dimensional space-time” and “biological clock” views of biological scaling are problematic, a time perspective based on time-sensitive biological responses to varying rates of mortality or destruction at various hierarchical levels of organization (e.g., cell, tissue, organ, organism, population, and species levels) has much potential for greatly increasing our understanding of why the rates and durations of various biological processes vary both with and independently of organismal size the way that they do (Figure 3). Scaling living systems in relation to their extent (existence) in not only space, but also time, may also significantly advance our understanding of biological scaling.

## 4. Appendix: What Is “Allochrony”?

In this essay, I chiefly use the term “allochrony” to refer to analyses examining how relatively short biological time periods scale with longer time periods (Ref. [138]; also see Section 2.3), thus paralleling how the term “allometry” is often used to designate analyses examining how the magnitude of a specific structure or process scales with body size. Smith [137] also used the term “allochronic” to describe any disproportionate relationship between different life-history time periods (loglinear slope ≠ 1). If the scaling relationship between two different biological time periods is proportionate (loglinear slope ≈ 1), it is called “isochronic”, in a similar way that the proportionate relationship between the magnitude of a structure or process and body size is called “isometric”.

However, the reader should be aware that the terms “allochrony” and “allochronic” have been used in other ways in different biological contexts. For example, evolutionary biologists have used the word “allochronic” to indicate that an evolutionary change in breeding time has contributed to the reproductive isolation between two species [282,283]. Recently, some developmental biologists have also described changes in the rates of specific developmental processes as “developmental allochrony” [284,285,286]. This term partially overlaps with the more general term “heterochrony”, which refers to a dissociation between different developmental processes through changes in either their rates or initiation times [136,144,285,286].

## Figures and Tables

**Figure 1 biology-12-01084-f001:**
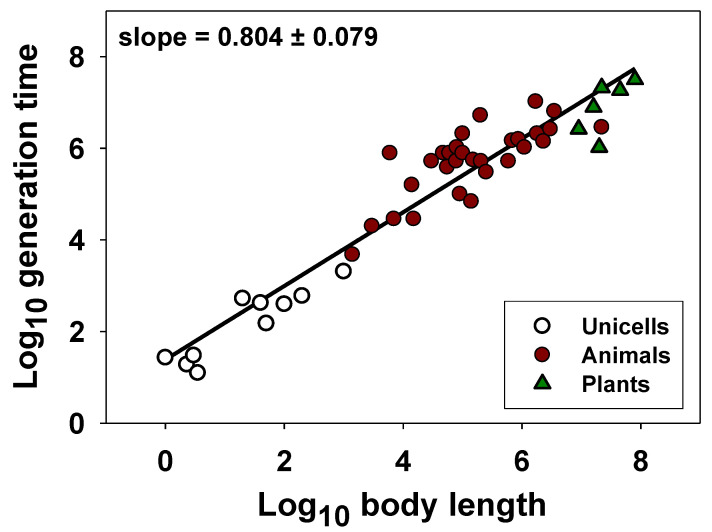
Generation time (approximated as age at first reproduction, minutes) in relation to length of the organism (µm), both log_10_ transformed to show proportional relationships (data from [116]). Unicells include prokaryotes and protists. Animals include invertebrates and vertebrates. Plants include kelp and gymnosperm and angiosperm trees. The scaling slope (±95% confidence intervals) for the linear regression line is indicated. Statistical details are provided in Table 1.

**Figure 2 biology-12-01084-f002:**
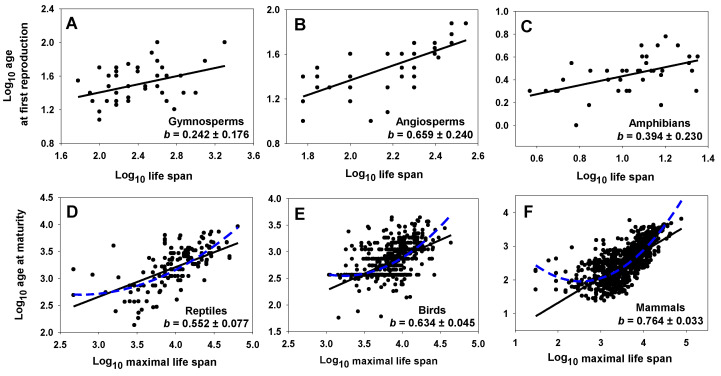
Age at maturity or first reproduction in relation to total life span, all log_10_ transformed to show proportional relationships. Scaling slopes (±95% confidence intervals) for the linear regression lines are indicated. Statistical details and data sources are provided in Table 2. (**A**–**C**), Age at first reproduction in relation to life span, both in years, for gymnosperm and angiosperm trees and amphibians. (**D**–**F**), Age of female maturity in relation to maximal life span, both in days, for reptiles, birds, and mammals. The blue dashed lines indicate quadratic (polynomial) relationships: reptiles: Y = 4.946 − 1.632(X) + 0.296 (X^2^) (*r* = 0.722, *N* = 223, *p* < 0.0001); birds: Y = 8.585 − 3.735(X) + 0.578 (X^2^) (*r* = 0.660, *N* = 1095, *p* < 0.0001); and mammals: Y = 4.720 − 2.184(X) + 0.432 (X^2^) (*r* = 0.770, *N* = 1793, *p* < 0.0001). For all curvilinear regressions, both the X and X^2^ terms are highly significant (*p* < 0.0001).

**Figure 3 biology-12-01084-f003:**
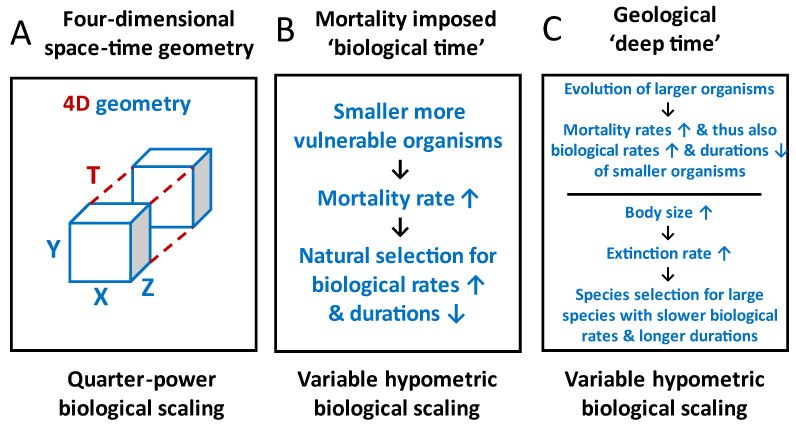
Three explanations of biological scaling based on a time perspective. (**A**), A “four-dimensional space-time view” assumes that time is a fourth dimension that is commensurate (proportionate) with the three dimensions of space (as depicted by a 3D cube moving through 1D time). Some investigators have claimed that this 4D view explains quarter-power scaling of the rates and durations of various biological processes. (**B**), A “biological time perspective” views size-dependent mortality rate as a driver of the body-mass scaling of the rates and durations of various biological processes, where the slope is typically <1 (hypometric) in log–log space. (**C**), A geological “deep time perspective” considers how the evolution of larger organisms may have affected the mortality rate of smaller vulnerable organisms and, thus, the rates and durations of their biological processes. This large-scale evolutionary perspective also includes the hypothesis that species selection on large species highly vulnerable to extinction because of their relatively small population densities may have favored relatively low rates of resource use in these species so as to increase resource availability per individual. Both of these hypotheses predict that the body-mass scaling slope for the rates and durations of various biological processes should be <1 (hypometric) in log–log space. See text for further details.

**Table 1 biology-12-01084-t001:** Statistical parameters for least squares linear regressions between log_10_ generation time (G = generation time; AM = age at maturity; AFR = age at first reproduction) and log_10_ adult body length.

Taxon	G/AM/AFR (Units)	Body Length (Units)	Slope (±95%CI)	Intercept (±95%CI)	r	N	*p*	Source
Unicellular and multicellular organisms	AFR (minutes)	µm	**0.804** (±0.079)	1.384 (±0.394)	0.953	46	<0.0001	[116]
Unicells	AFR (minutes)	µm	**0.731** (±0.249)	1.178 (±0.403)	0.920	10	<0.0001	[116]
Animals	AFR (minutes)	µm	**0.607** (±0.194)	2.591 (±1.020)	0.771	30	<0.0001	[116]
Cladocerans	AFR (days)	mm	**0.479** (±0.289)	0.791 (±0.078)	0.740	13	0.004	[119]
Teleosts	AM (years)	cm	**0.799** (±0.354)	−0.925 (±0.614)	0.791	16	<0.0001	[120]
Squamates	AM (months)	mm	**0.313** (±0.084)	0.595 (±0.178)	0.521	145	<0.0001	[121]
Reptiles	AM (days)	cm	**0.292** (±0.138)	2.478 (±0.183)	0.441	76	<0.0001	[122]
Birds	AM (days)	cm	**0.694** (±0.080)	1.656 (±0.135)	0.629	442	<0.0001	[122]
Mammals	G (years)	mm	**0.817** (±0.153)	−1.832 (±0.433)	0.904	29	<0.0001	[106,122,123,124]
Mammals	AM (days)	cm	**0.674** (±0.035)	1.487 (±0.055)	0.664	1815	<0.0001	[122]

CI = confidence intervals, *r* = Pearson product–moment correlation coefficient, *N* = sample size, *p* = probability that r is due to chance. Bold slope values are significantly less than 1.

## Data Availability

Not applicable.

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
