# Peer review of "The Relevance of Time in Biological Scaling"

_biology, 2023, doi:10.3390/biology12081084_

Round 1
Reviewer 1 Report
In this manuscript, the author proposes that time is much more than the fourth dimension of life history variation but as much and maybe even more than body size, timeplays a crucial role in shaping life history strategies.
The author asks for increasing allochronic analyses of life history traits in addition to allometric analyses and sets the base for a "mortality theory of ecology" to complement the current "metabolic theory of ecology".
I really enjoyed reading that nice and deeply informed opinion paper and has to say that I am fully convinced that the structuring effect of time has been too much neglected up to now in life history analyses and needs to be investigated much more thoroughly.
That being said, I found a series of major problems with the manuscript as it stands, all of them being quite easy to solve with a careful rewriting.
1. The evidence supporting the existence of a deviation between estimated scaling coefficient of the allometry between age at first reproduction and body length and expected values under the 4D space-time theory has to be poundered because (1) the linear regression used assumes independent data points, which is not the case of species studied here because of the existence of the principle of phylogenetic inertia (see Felsenstein 1985 Am. Nat. for a pioneer work) and (2) both age at first reproduction and body length are measured with errors and maybe even biases. Both of these issues lead to increase the standard error of the slope and make the deviation from the expectation less clear than stated at the best.
2. I fully share the author's view that "residual variation that is orthogonal to a body-size regression can be substantial" but the connection with the problem of scaling is far from clear. While it is (rightly) stated that "lifespan can vary by over 10-fold in mammals of equivalent size", this variation is not accounted for only by variation in the scaling coefficient. In particular, the intercept varies a lot. Consider bats and squirrels. In both groups lifespan increases with size with quite similar slopes, but the intercept for bats is much higher than the intercept for squirrels. That intercepts vary a lot and much more than the scaling coefficient has been noticed since a long time in comparative analyses of life history traits (see e.g. Western 1979 Afr. J. Ecol.) and has more recently be supported by evolutionary studies (see e.g. Bolstad et al. 2015 PNAS). This needs to be acknowledged.
(3) I have some concerns about interpreting life history variation as driven by the "pace of death" and see no justification of prioritizing this "pace of death" over a "pace of reproduction". The "pace of life", corresponding across species to the slow-fast continuum (after Stearns 1983 Oikos) integrates all the time components of the life cycle and there is no reason to identify a unique main driver. What drives life history can be context-dependent. For instance, the author writes rightly on l. 495-496: "If the mortality rate in a population of organisms increases, natural selection should favor a quicker rate of reproduction (and overall pace of life)", but stating that "If the reproductive rate in a population of organisms increases, natural selection should favor faster mortality (and overall pace of life)" is also fully correct. In the current period of climate warming, there is some empirical support to this second scenario. Thus, Dupoue et al. (2022 PNAS) found that higher body growth rates led to earlier reproduction and then to accelerated senescence, leading the pace of life to be faster. A more balanced presentation is thus required.
(4) The second answer provided in Box2 is unfortunately totally wrong. There is no programmed evolution that would lead each individual in a population to replace itself at the next generation. The theory shared by the community of evolutionary ecologists involves the process of natural selection as defined initially by Darwin (1859) and formalized by Fisher (1930), which leads to maximise (not optimize!) the individual fitness that corresponds to the contribution of a given individual to future (in most real cases the next) generations. The huge and increasingly reported individual heterogeneity in fitness within populations support a process of maximisation instead of optimization. So the interpretation (as far as I understood it, if not please explain) that mortality is the driving process to adjust reproduction and ultimately growth and metabolism is flawed.
Detailed problems:
There is an overuse of the term "intrinsic and extrinsic factors", whose meaning is unclear at the best. In the real word, both types of factors strongly interact, which leads the distinction between these types of factors useless.
l. 207-208 and Figure 1 legend: Generation time, which corresponds to the mean age of mothers in a population, cannot be "estimated" as the age at first reproduction. At the best, age at first reproduction provides a proxy for generation time, as both traits are quite closely related (see e.g. Gaillard et al. 2005 Am. Nat.)
l. 496-498: Please rewrite. This stands as an unnecessary programmed evolution argument. Simple life history theory based on natural selection theory and principle of allocation explains that easily in absence of any global plan of evolution.
l. 498-500: This is a very naive view that is not supported by empirical data demonstrating the increasing frequency of both invasive species that combine high survivorship and high reproductive output to maintain fast population growth rates and extend their spatial distribution and declining species that combine low survival and low reproductive output to decrease in abundance up to extinction.
l. 500-503: This sentence is confusing at the best. While it is true that the "rate of living" theory has been criticized as well illustrated by the three references provided in support, the association among faster senescence, reduced longevity, and higher mortality does occur. This association that corresponds to the second prediction of Williams' seminal work (1957 Evolution) has been repeatedly supported at the interspecific level treated in the present work (see Gaillard & Lemaitre 2017 Evolution for a review).
l. 506: I cannot agree more with this statement that echoes with my comment above about the partition of the pace of life into pace of death and pace of reproduction (and certainly also developmental pace). The text has to be restructured and altered to reflect this view and not overemphasize the role of the pace of death as it currently does.
l. 529: "are associated" does not mean "are driven by"!
l. 532-533: Again, natural selection is not aimed to reach any "ecological compensation". It is a biological process that has no aim at all (at least we don't need to attribute an aim to natural selection!). If ecological compensation was the rule on earth, we should not observed so many invasive or declining populations.
Author Response
Thank you for your support. I have responded to each of the reviewer’s comments below.
- I am looking for general trends (slopes < 1), which are observed in every taxon. Including phylogenetic effects may alter the exact value of the exponent, but are unlikely to alter this general conclusion. Please also be aware that when I subdivided part of the data set of Bonner (1965) into protists and animals, negative allometry of G vs. L was still observed for each group, showing that such patterns are unlikely to be phylogenetic artifacts. Inspection of Figure 1 reveals that the generally negatively allometric pattern is unlikely to be due to phylogenetic effects, as well. In addition, note that body length is typically measured with less error than generation time, and thus use of least squares regression is appropriate.
-
Here I am not attempting to explain variation in the scaling exponent among different regression lines, but rather pointing out that much of the variation in life span is not explained by variation in body size within each group of species for which a scaling relationship has been calculated. The reviewer does not distinguish residual variation around a regression line that I focus on with variation in intercepts among regression lines.
-
Elsewhere in my paper I already mention that mortality rate may be both a cause and result of the pace of life, as described by the ‘rate of living theory’. In my review, I emphasize that body-size related mortality rate may importantly drive the body-size scaling of the pace of life, because this view has been neglected. It is important to realize that reproduction would be unnecessary if mortality did not occur. Natural selection favors (faster) reproduction to offset (faster) mortality.
-
To clarify, my argument is that natural selection favors faster growth/reproduction (pace of life) to offset a faster mortality rate. That is all. I do not argue for any kind of “programmed evolution”.
Responses to detailed comments follow in the order that they were presented.
I respectfully disagree. Intrinsic factors refer to biological properties of the organism itself, whereas extrinsic factors refer to environmental factors (as I have clarified in the manuscript). That they interact does not negate this useful distinction.
Yes, that is what I meant. This has been clarified.
I propose no “global plan of evolution”. Mortality rate is considered a driver that is mediated by natural selection.
Increasing or decreasing population growth rates are necessarily temporary. In the long run, mortality and reproductive rates must balance if a population is to persist. Population growth of invasive species eventually declines because of resource limitation or other limiting factors, whereas continually declining populations eventually disappear.
I agree in large part with this comment, which is why I stated that reciprocal effects can occur. Note that Gaillard & Lemaitre (2017) only consider the theory of senescence of Williams (1957), which supports the view that mortality rate drives the pace of life (including the rate of aging) as is emphasized in my review. Since they do not explicitly discuss the rate of living theory that metabolic rate drives the pace of life and death, I did not add this reference here.
I consider the pace of death as separate from the pace of life (which includes the pace of metabolism, growth, development, and reproduction) and as a driver of the pace of life only via the action of natural selection. After all, death is the absence of life.
Yes, but these expressions are not antithetical, as I have clarified in the manuscript. Such associations provide support for the idea that rates of mortality/destruction may drive the rates of various biological processes, which, of course, requires further testing.
To persist in the long run, populations cannot grow/decline indefinitely. Ecological compensation must operate to ensure long-term population persistence (stability).
Reviewer 2 Report
Based on empirical and logical evidences, this paper refuted the universal four-dimensional space-time and ‘biological clock’ views of biological scaling, and advocated using a time perspective in three other major ways to increase understanding of many kinds of biological scaling patterns. This paper also recommended some very interesting topics for future research. I think this paper has provided a very comprehensive review on previous studies and very meaningful suggestions for future studies. I only have three questions that might need the author to address: 1) in part 2.2.1, it says that there are many organisms that do not follow the quarter-power scaling, therefore we should not use this simple 4D view to explain. But there are also many organisms whose body-mass scaling of the rates and durations of processes show quarter-power relationship, how should we explain such consistency in so many organisms? Also, there seems to be few theories that can be applied to all organisms. 2) line 226, it seems that it was not explained in the text what M meant. 3) in line 602-609, I am not sure about the reliability of this part. For example, the first multicellular might be benthic, not good at preying, and eat organic matters from sediments. They should be in different ecological niche and thus might not compete with unicellular organisms.
Author Response
Thank you for your support. Responses to the reviewer’s 3 points follow:
1) Please refer to Glazier (2022) PRSB, where it is shown that diversity of metabolic scaling (not consistency) is the dominant pattern. The 4D view cannot easily explain this diversity without significant modification (as I state in the manuscript).
2) Thank you for noticing this. I have now defined M in lines 226-227.
3) I have taken care of this problem by the rewording: “many multicellular metazoans and protists that evolved early in the history of life” on lines 620-621. Note that some of the earliest multicellular metazoans, such as sponges, ctenophores and cnidarians, were clearly predators.
Reviewer 3 Report
This is a thought-provoking review article that proposes to add time as a factor in explaining biological scaling. The article is well-written. I have two minor points:
1. One complication about metabolism rate is that, recently, it has been suggested to be relevant for regulating aging, longevity and developmental timing. Although this still needs to be tested more vigorously further, one indication from this is that it is not easy to separate "time" from metabolism rate in the life-history of an organism. What is the author's thought on this?
2. Items in table 1 and table 2 need to be aligned. Some columns look a bit messy.
Author Response
Thank you for your support. Responses to the reviewer’s 2 points follow:
1) Actually, the possible effect of metabolic rate on aging, longevity and developmental timing is part of the “rate of living” theory, which I address in the manuscript on L 505-512: “although many biologists have embraced the ‘rate of living theory’ that a quicker pace of life (including metabolic rate) causes quicker aging, reduced longevity and ultimately a higher mortality rate (e.g., [26,56,203-206]), the generality of this view has been questioned (e.g., [28,207,208]). I argue that the opposite causation where mortality rate drives the evolution of the rate of reproduction and overall pace of life (as promoted by [124,199-201,207,209,210]; and others) has also been significant. Indeed, both types of causation may be reciprocally important.”
2) Unfortunately, tables 1 and 2 were altered after submission of my manuscript. I have now fixed them.